# Progress in the Realization of μ-Brush W for Plasma-Facing Components

**Daniel Dorow-Gerspach** [1,*], **Thomas Derra** [2], **Marius Gipperich** [2], **Thorsten Loewenhoff** [1], **Gerald Pintsuk** [1], **Alexis Terra** [1], **Thomas Weber** [1], **Marius Wirtz** [1] **and Christian Linsmeier** [1]

1   Forschungszentrum Jülich, Institut Für Energie- und Klimaforschung—Plasmaphysik, 52425 Jülich, Germany
2   Fraunhofer-Institut Für Produktionstechnologie IPT, 52074 Aachen, Germany
*   Correspondence: d.dorow-gerspach@fz-juelich.de

**Abstract:** During the service life of plasma-facing components, they are exposed to cyclic stationary and transient thermal loads. The former causes thermal fatigue and potentially detachment between the plasma-facing material tungsten and the structural Cu-based materials (divertor) and steel (first wall). The latter causes surface roughening, cracking, or even melting, which could drastically increase the erosion rate. Employing thin flexible W wires ($W_w$) with a diameter of a few hundred μm can reduce mechanical stresses, and we demonstrated their crack resilience against transient loads within first proof of principle studies. Here, status and future paths towards the large-scale production of such $W_w$ assemblies, including techniques for realizing feasible joints with Cu, steel, or W, are presented. Using wire-based laser metal deposition, we were able to create a homogeneous and shallow infiltration of about 200 μm of the $W_w$ assembly with steel. A high-heat-flux test on such a μ-brush ($10 \times 10 \times 5$ mm³ $W_w$ on a ~0.5 mm thick steel layer) using 5 MW/m² for 2000 cycles was performed without loss of any wire. Microstructural examination after and infrared analysis during the test showed no significant signs of degradation of the joint.

**Keywords:** μ-brush; plasma-facing armor; tungsten; cyclic thermal load; tungsten steel joint

## 1. Introduction

Tungsten (W) is used in a variety of applications that benefit from its unique properties such as having the highest melting point and lowest vapor pressure of all metals, as well as high thermal conductivity, strength, and chemical resistance [1]. Among other things, because of these properties it is the main candidate for the armor of plasma-facing components (PFCs) in thermonuclear fusion reactors [2] and other applications where the material has to withstand very high thermal loads at low erosion rates, such as rocket nozzles [3] or anodes for X-ray tubes [4]. However, the high brittle to ductile transition temperature and poor machinability brings the necessity of joining it to a structural material, which also includes the cooling circuit. Cu or CuCrZr are the main candidates for this in the divertor, which has to sustain the highest steady-state heat loads of up to 20 MW/m² in ITER (International Thermonuclear Experimental Reactor) or even more in future reactors, whereas due to lower cost and activation, special low activating ferritic/martensitic steels (e.g., EUROFER97) are envisioned for the PFC heat sink material of the first wall [5], which covers the main part of the reactor but has to sustain only about 1 MW/m². As W has a much lower coefficient of thermal expansion (CTE) than Cu and steel, thermal stresses occur at the respective interfaces, which can lead to a failure of the joint due to cyclic fatigue, especially taking embrittling effects such as neutron irradiation, hydrogen, or helium incorporation into account. In this paper, a possible path to solve this challenge is presented by joining thin W-wires ($W_w$), instead of bulk W, to steel.

Another critical aspect for a PFM is its behavior under transient heat loads, such as type 1 edge localized modes (ELMs), having an intensity in the order of GW/m² on a ms

time scale, originating from plasma edge instabilities [2]. These high heat loads lead to cracks in W in the long term, massive surface deformations, and local overheating and melting [6]. Previously, we demonstrated, in a proof of principle study, that flexible $W_w$ are resistant against these loads [7]. However, the manufacturing process involved short pre-sliced $W_w$, which had to be ordered manually to a closely packed assembly before infiltration by molten copper in a graphite mold. In a next step, most of the Cu had to be etched out again to obtain free-standing $W_w$ that were joined to Cu at one end only. The very high aspect ratio (gaps between wires < 50 µm vs. length of the $W_w$ of a few mm) and the large capillary forces due to the good wetting of W with Cu makes the manufacturing of a reproducible and homogeneous Cu infiltration level very challenging. The lack of scalability of the manual approach and the need for a robust, uniform, and reproducible joining technique prompted the search for other manufacturing procedures. Several possible techniques have been studied, and the currently most promising ones to join a µ-brush of $W_w$ with Cu, steel, or W, as the most relevant possible joining materials, are presented in this paper.

## 2. Materials and Methods

### 2.1. Creating a Versatile $W_w$ Assembly

The basic idea for the development of an up-scalable µ-brush was to find a procedure to create a stable $W_w$-assembly with a high degree of freedom in the choice of the length of the wires and the joining technology. It should also be based on standard techniques to enable rapid adaptation by industry later on and at the same time ensure high quality (in particular, the high density of the packed assembly). The theoretical maximum density can simply be calculated by considering the topview of the densest possible packing and an equilateral triangle, as shown in Figure 1.

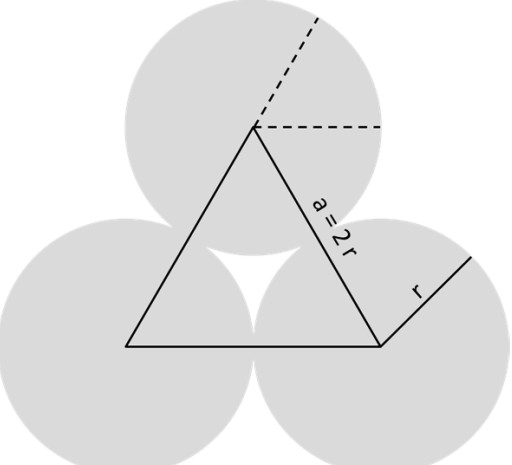

**Figure 1.** Equilateral triangle to determine the maximum density of a packed wire assembly.

The volume of a triangular pillar and a wire are both determined by area of the face, triangle, and circle, respectively, times their length l. The area of an equilateral triangle is $A_t = \frac{1}{4}a^2\sqrt{3}$ (with *a* being the length of the sides) and of a circle $A_c = \pi r^2$ (with *r* being the radius). Considering the relation shown in Figure 1, the theoretical achievable density $d_{th}$ is determined by

$$
\begin{aligned}
d_{th} &= \frac{3 \times \frac{1}{6} \times A_c \times l}{A_t \times l}, \\
&= \frac{1}{2}\frac{\pi}{\sqrt{3}}, \\
&\approx 90.7\%
\end{aligned} \tag{1}
$$

The maximum density, which can be achieved by densely packing wires, is approximately 90.7% independent of the chosen diameter and length. In reality, the stacking will not always be perfect and thus the density a bit lower.

Machine winding on a square (76 × 76 mm$^2$) spool (Figure 2a) was used to achieve high density and as it is in principle an up-scalable procedure, similar to the production of electromagnets. Stainless steel was chosen for the material of the spool as it is easy to machine and can sustain high temperatures, allowing for a larger variety of joining techniques later on.

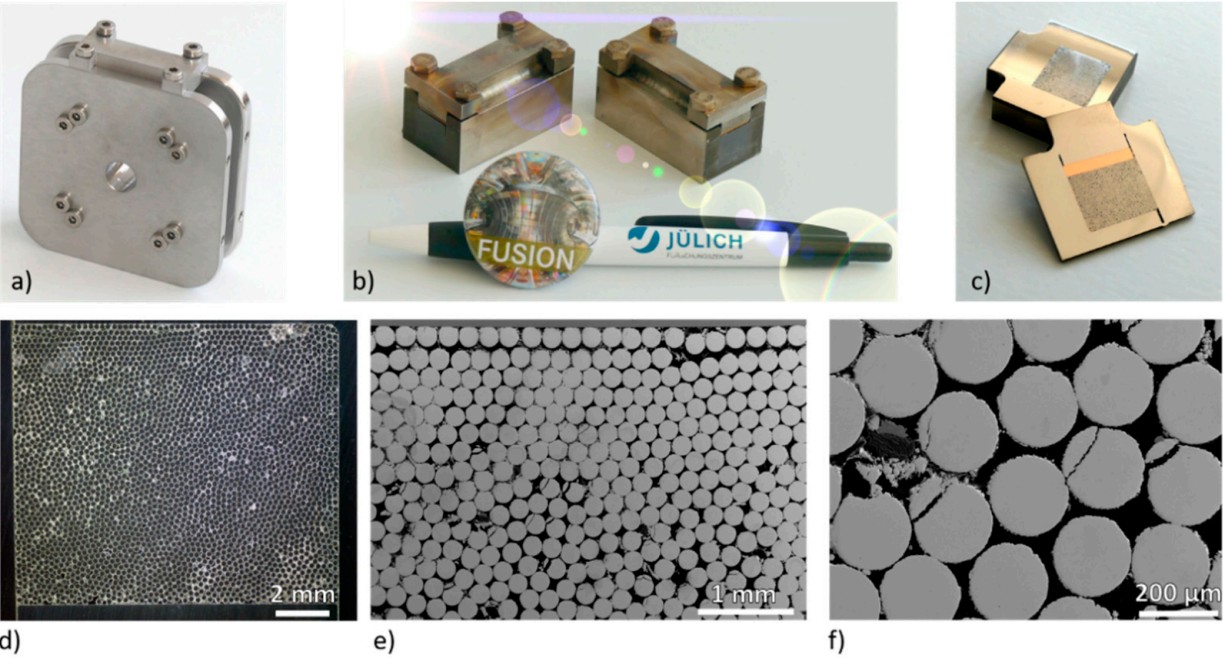

**Figure 2.** Manufacturing procedure is based on sectioning a spool (**a**) after winding the W-wire around and welding on clamps at the edges. The sections are shown in (**b**) and the surface of a typical $W_w$ slice after polishing in (**c**–**f**).

We used drawn, potassium-doped $W_w$ purchased from OSRAM GmbH (Schwabmünchen, Germany), named BSD, with a diameter of 200 μm. Lower diameters were tested as well (50 and 100 μm) but did not display a distinct advantage. For a 12 × 12 mm$^2$ cross section, about 4000 windings were necessary; thus, this was done by a specialized company, the M.Richter GmbH and Co.KG (Pforzheim, Germany). Next, clamps were attached at the edges and welded to the spool. Afterwards, the spool could be divided by electro discharge machining (Figure 2b) and then sliced and polished to the desired thickness respectively wire length (Figure 2c). In Figure 2d–f, images of the resulting wire assembly are shown. Image analysis showed that a density of up to 85% could be achieved. However, quite a few stacking faults are still present, and some wires were damaged at the tip by the slicing and grinding; thus, the winding and surface-preparation process needs to be improved in the future.

### 2.2. Creating a μ-Brush

After the slicing and polishing were successfully carried out without losing the integrity of the wire assembly, a μ-brush could be manufactured by joining it to the desired base material depending on the envisioned field of application. The first experiments were performed by infiltrating it with Cu in a graphite mold followed by electro chemical etching to obtain free-standing wires [7]. However, this technique is not well suited to the large-scale production of a firm but shallow, reproducible, and homogeneous Cu level

across the whole sample. Therefore, we tried a variety of alternatives to realize the joint between $W_w$ and the three most important materials for PFCs: Cu, steel, and W.

In Figure 3, the three most promising techniques so far are illustrated. For Cu, field-assisted sintering, also called spark-plasma sintering, shows promising results, using sintering parameters of 800 °C, 50 MPa for 5 min, and a Cu powder of particle size < 45 μm. However, further optimization is needed as the joint is established at the top face of the wire only and is thus predictably weak considering the small contact surface.

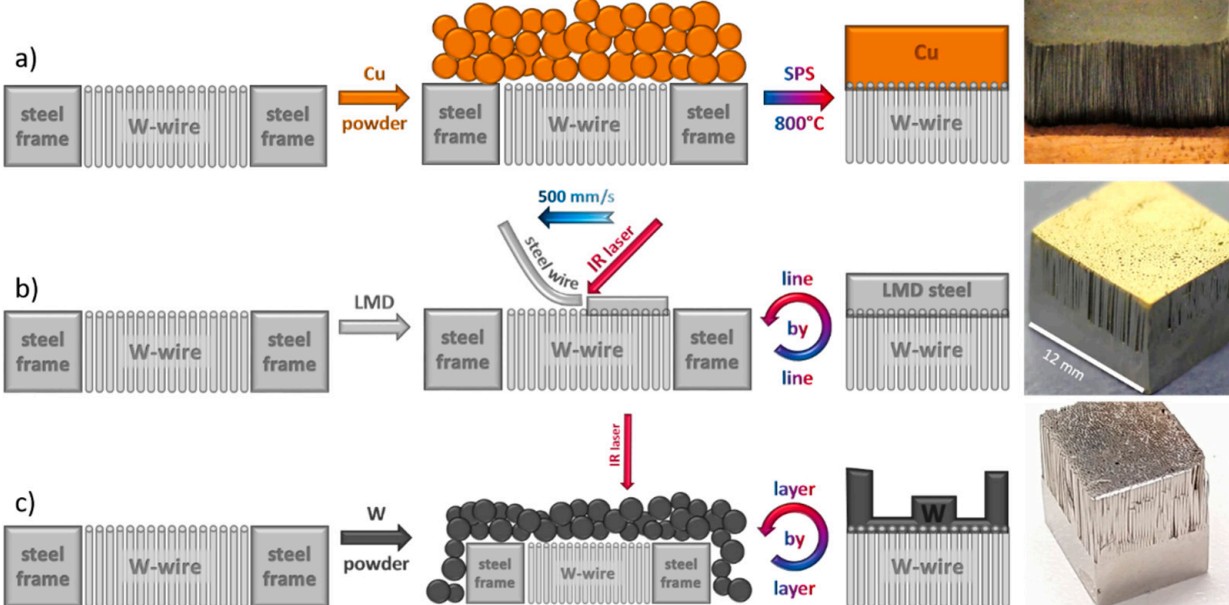

**Figure 3.** Developed joining techniques to create a μ-brush with high potential for up-scaling in the future. For Cu spark plasma sintering (**a**); for steel laser metal deposition (**b**); and for W selective laser melting, which also enables advanced structures (**c**).

To realize the W-steel μ-brush, a very promising process was developed, using the wire-based laser metal deposition at the Fraunhofer IPT in Aachen, Germany (illustrated in Figure 3b). It uses an Ar gas flow to protect the heated area from oxidation, and a 1 mm-thick steel wire is fed to the substrate surface, which gets heated by a continuous diode laser. The process parameters are denoted in Table 1, and a low-alloyed steel (Böhler DMV 83-IG from BÖHLERWELDING (Düsseldorf, Germany) was used for the deposition. More technical details can be found in [8,9].

**Table 1.** LMD deposition parameters for W-steel μ-brush.

| Parameter | Value |
| --- | --- |
| Laser power | 1400 W |
| Travel speed | 500 mm/s |
| Wire feed rate | 550 mm/min |
| Ar gas flow | 12 L/min |
| Temperature | Room temp. |

The advantage of this process is that it melts the steel only locally and very briefly, and thus the infiltration depth can be controlled. It is in the range of 200 μm, as can be seen in the light microscope image of a cross section in Figure 4. In addition, the deposited layer and the infiltration depth are homogeneous across the whole sample. The joint is very robust, as indicated by the fact that the sides of the wires can be partly ground without becoming loose and by the fact that cross sections can be made.

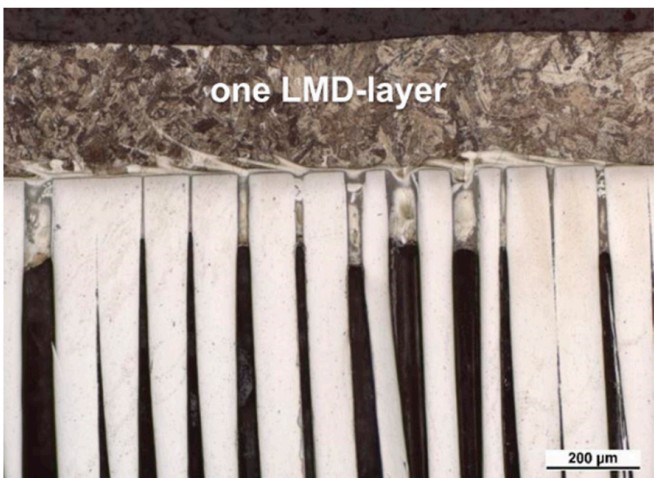

**Figure 4.** Light microscope image of a cross section of W-steel μ-brush prepared by LMD.

We also tried to use the same technique for the W-W μ-brush; however, it turned out that the much higher temperatures necessary for melting W led to a large degree of porosity and bubbles within the LMD-layer and a very wavy surface; thus, LMD seems to be not very well suited for this combination.

Therefore, it was tested to apply the process developed by the Fraunhofer IGCV, together with the Max Planck Institute for Plasma Physics (IPP Garching), for printing W by selective laser melting (SLM), a powder bed-based additive manufacturing (AM) technique, on the $W_w$-assembly (Figure 3c), see also [10]. Using AM would allow the realization of advanced and individually designed structures without elaborate machining. Indeed, the first samples look quite promising, as the joint was firm enough to allow grinding the sides and the AM-W showed practically no porosity. Thus, the combination of SLM of W and afterwards LMD of steel on the other face of the $W_w$ could allow for an interesting new type of W-steel joint in the future.

*2.3. High-Heat-Flux Tests*

The samples were soldered to actively cooled Cu cooling structures using an Ag–Cu braze, allowing for the efficient removal of applied heat and thus a higher frequency during the thermal shock loading tests. The electron beam facility JUDITH 2, located at Forschungszentrum Jülich, was used for these tests using specifically designed loading patterns for high-pulse-number transient thermal loads at a high frequency (25 Hz) [11]. For the cooling, a water temperature of 70 °C, a pressure of 20 bars, and a flow velocity of about 25 m/s were employed. The surface temperature was monitored using an IR camera and two-color pyrometers. A stationary heat load was applied prior to the transient heat load tests to achieve a base temperature of 700 °C. The additional transients had an intensity of 0.55 GW/m$^2$ and a duration of 0.48 ms (heat flux factor of $F_{HF}$ = 12 MW/m$^2$s$^{0.5}$). For the cyclic steady-state heat load test, 5 MW/m$^2$ was applied on three W-steel μ-brushes with different $W_w$ lengths of 5, 6.4, and 10 mm to illustrate the variability of the manufacturing technique. During each cycle, the beam was 30 s on and 30 s off, and a total of 2000 cycles were performed.

**3. Results**

The surface and cross section after the transient heat load experiment can be seen in Figure 5. As a reference, a bulk W manufactured according to ITER specifications by Plansee SE (IGP-T) was included. After 10$^5$ pulses, the IGP-T surface is massively deformed, including local melting and severe irregular crack formation, which is not the case for the μ-brush W either after 10$^5$ or after 10$^6$ pulses.

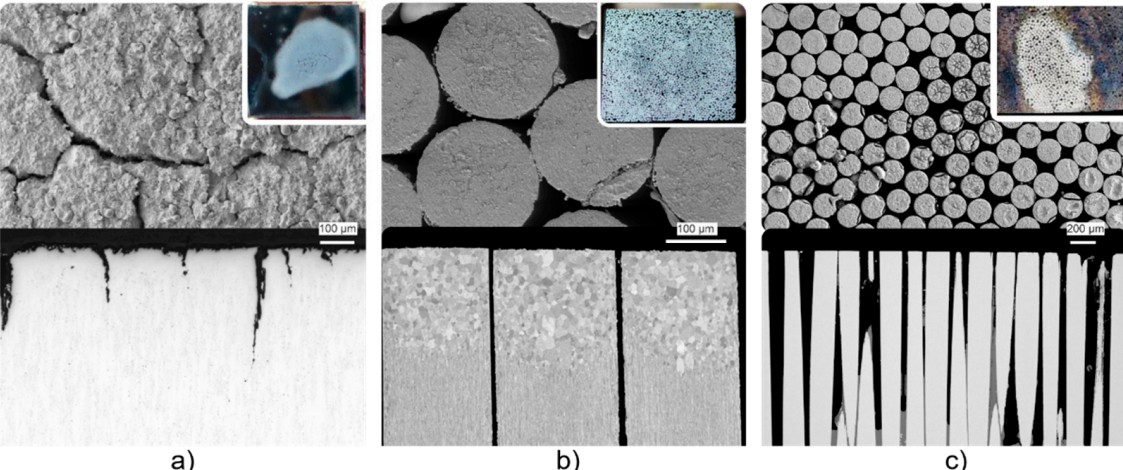

**Figure 5.** Surface (upper) and cross section (lower) of bulk W reference after $10^5$ transients (**a**) and W-Cu μ-brush joined by Cu infiltration after (**b**) $10^5$ and (**c**) $10^6$ transients. The insets show a top-view picture of the whole sample.

The recrystallization visible at the tip of the μ-brush shown in the middle of Figure 5 is due to an unintentionally higher stationary loading during the setup prior to the transients, resulting in a surface temperature of about 1600 °C for about half an hour. Despite this recrystallized structure, which is known to have very brittle wires [12], no cracking was caused in the W by the transients. It should be noted that the fine cracks at the surface are only present in a nm thin CuO deposit, as was confirmed by energy-dispersive X-ray spectroscopy (EDX), originating from sublimated Cu, showing the necessity of an alternative for infiltrating with Cu, as was mentioned earlier. The tips of some wires had been already damaged during the grinding and polishing, as can be seen in Figure 2, and they differ from the crack appearance caused by transients.

To further validate that the wires were not damaged by the transients, a direct comparison of the top-surface after manufacturing and after the transient heat load is given in Figure 6. It is the same sample as in Figure 5b, and as can be seen, the broken wires had been damaged already during the manufacturing but were not altered by the $10^5$ transients. The remaining W-debris has of course only limited if any heat-removal capability, which explains the few molten droplets at the edge of some wires, after the loading.

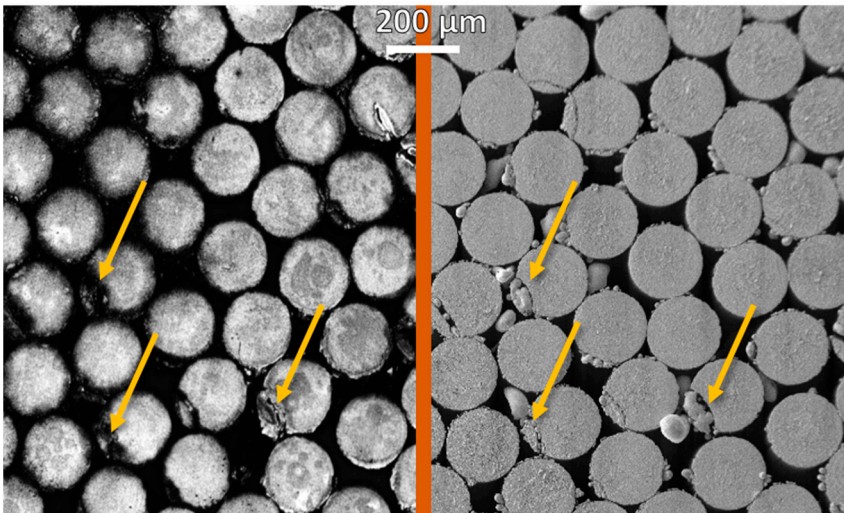

**Figure 6.** Comparison of the surface of the μ-brush after manufacturing (light microscope: **leftside**) and after $10^5$ transients (SEM image: **rightside**). All broken wires were damaged before the high-heat-flux loading (e.g., marked by arrows).

During the cyclic steady-state heat loads, the surface temperature was between 600 °C and 900 °C depending on the length of the $W_w$. No wire became loose, and the surface temperatures did not change significantly during the cycling, indicating a stable bonding between W and steel. Figure 7 shows an SEM image of the cross section of the interface region, including the positions of the EDX measurements, whose results are given in Table 2. FeW intermetallics formed below and between the $W_w$, but no cracks or bond failures were detected. The basic fact that cross sections could be made without breaking is a remarkable result as this could not be done with diffusion-bonded bulk W and steel [13]. Thorough investigations and test series are planned to investigate the grow rate of the zone of intermetallics and the long-term stability.

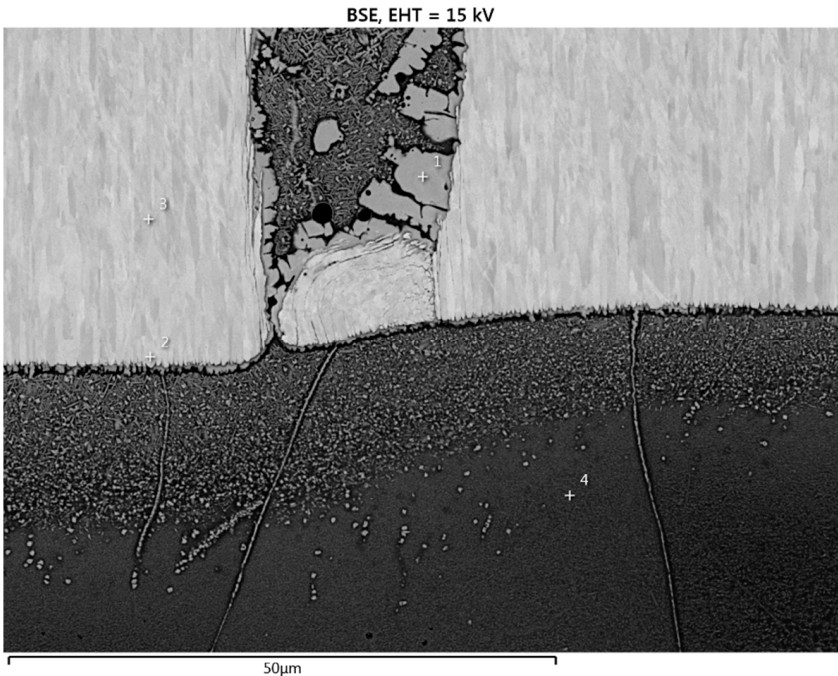

**Figure 7.** SEM image of the cross section at the interface of $W_w$ and steel after 2000 cycles with 5 MW/m$^2$. The numbers indicate the locations of the EDX measurement spots.

**Table 2.** Local composition (labeled in Figure 7) determined by EDX in at%.

| Label | Fe | W |
|---|---|---|
| Position 1 | 57.8 | 42.2 |
| Position 2 | 5.4 | 94.6 |
| Position 3 | 0 | 100 |
| Position 4 | 97.6 | 2.4 |

## 4. Discussion

The resilience of thin $W_w$ against transient heat loads had been demonstrated before [7] and could also be shown for the W-Cu μ-brush produced here by Cu infiltration and chemical etching. The fact that even after recrystallization, the μ-brush remained resistant against transient heat loads is excellent news for the long-term stability of this material concept.

The correct wire diameter is of vital importance for the presented approach. In the past, castellated tungsten, called macro-brush, was considered as plasma-facing material for the divertor of ITER [14] or as a limiter [15] with the dimensions of the square "tiles" in the range of 1 × 1 cm$^2$. Experiments with ELM such as loads resulted in the cracking of the W similar to bulk W [14]. The typical crack distance in bulk W resulting from such transient heat loads is in the range of 0.4 mm to 0.6 mm, irrespective of the intensity or

used energy source (e.g., laser, ebeam) [16]. Therefore, the diameter of the wire has to be lower to cope with the evolving stresses, and we mainly used 200 μm as even smaller ones down to 50 μm were available but would increase the fabrication costs without any clear benefit. Additionally, the macro-brush concept has continuous poloidal and toroidal edges due to the gap width of 0.3 to 0.5 mm; they are, therefore, prone to overheating and melting. However, the hexagonal pattern of the circular wires of the micro-brush concept, also described here with much smaller gaps (<50 μm see Figure 2f), distributes the heat loads better and should show less edge melting when loaded under grazing incidence.

It was also demonstrated by the unscathed survival of the high-cyclic-heat-load experiments that the concept of maintaining stress within the elastic regime by reducing the critical dimension, using thin $W_w$, allows for a stable bond between W and steel to be established despite their large CTE difference and could be a solution for long-term, stable W-steel joining technology, which is at the moment still missing [17].

It can be concluded that with the μ-brush concept, transient-resistance W-wires can be deployed in numerous areas of application, including as divertors, first walls, liquid walls, or limiters, by using different lengths and the developed bonding techniques to combine them with various structural materials. In addition, as it is based on more or less standard industry techniques such as winding, welding, and cutting, an affordable up-scaling should be possible.

**Author Contributions:** Conceptualization, D.D.-G. and T.W.; methodology, T.L., T.D., M.G. and A.T.; investigation, D.D.-G. and M.G.; resources, G.P. and C.L.; writing—original draft preparation, D.D.-G.; writing—review and editing, T.L. and G.P.; supervision, G.P., C.L. and M.W.; project administration, D.D.-G.; and funding acquisition, D.D.-G., T.L. and M.W. All authors have read and agreed to the published version of the manuscript.

**Funding:** This work has been carried out within the framework of the EUROfusion Consortium funded by the European Union via the Euratom research and training program (grant agreement no. 101052200—EUROfusion). The views and opinions expressed are, however, those of the authors only and do not necessarily reflect those of the European Union or the European Commission. Neither the European Union nor the European Commission can be held responsible for them.

**Data Availability Statement:** Not applicable.

**Acknowledgments:** We would like to thank Alexander von Müller from the Max Planck Institute for Plasma Physics (IPP Garching) and Thomas Bareth and Maximilian Binder from the Fraunhofer IGCV for adapting their SLM process of W for the use of $W_w$ as a substrate to create a W-W μ-brush.

**Conflicts of Interest:** The authors declare no conflict of interest.

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
