# Peer review of "Progress in the Realization of µ-Brush W for Plasma-Facing Components"

_jne, doi:10.3390/jne3040020_

Round 1
Reviewer 1 Report
The authors present novel fabrication methods and the characterization of the resulting tungsten micro-brushes in particular with respect to their thermal stability.
The work is timely and interesting. I believe that the research should be published in the Journal of Nuclear Engineering subject to the following minor amendments:
(1) Line 47: "... loads to cracks ..." should probably read "... loads lead to cracks ..."
(2) Figure 2: The image (b) is explained in the text, however not in the figure caption. This should be amended. Also the scale bars are very difficult to discern. It would be very helpful to improve this.
(3) Table 1: The authors write that more technical details on the LMD parameters can be found in the literature. However, it would be useful to state whether a "continuous" or a "pulsed" laser was used.
(4) Figure 5: There are insets to the upper three images in this figure. I presume that they show the original samples (i.e. not magnified). This needs to be clarified and stated in the figure caption and/or the text.
Author Response
We want to thank you for your positive assessment of our manuscript and your helpful comments in order to improve it.
(1) Changed accordingly
(2) Scale bars were changed and the caption amended.
(3) It was a continuous one, although an additional pulsed one is also available at the machine.
(4) Yes, they show a picture of the whole sample and this info was added.
Reviewer 2 Report
The manuscript entitled " Progress in the realization of µ-brush W for plasma-facing components" is well written and provides a unique method for fabricating a new concept of W brush. The detailed manufacturing process has been described and high heat flux tests have been carried out to check the structure stability. Minor revisions should be done before considering the manuscript to be published.
1. In line 161, three different lengths of 5, 6 and 10 mm have been used, but in the followed parts the differences leading from the length variation have not been discussed in detail.
2. In Fig.5 (b) and (c), two different scale bars have been used. So in Fig.5 (c), a larger field of view showed some broken fibers. How about the same position in the case of sample in Fig.5 (b).
3. Since the pure W sample in Fig.5 (a) has severe crack and local melting phenomenon, has there any recrystallization happened?
4. Since recrystallization happened before followed heat flux tests in sample of Fig.5 (b), the microstructure of the top layer of the fiber has changed in a fully recrystallized state. As we know, recrystallized structure has a lower thermal load resistance. Why no cracks have been found?
5. The section of conclusion should be given.
Author Response
Dear reviewer,
we would like to thank you for your positive assessment of our manuscript and your helpful comments in order to improve it.
- The length variation was only mentioned in order to illustrate that three samples have been tested and all of them survived the high heat flux tests without the loss of any wire. The length itself has no influence on the bonding but shows the variability of the manufacturing approach and that the length can be chosen freely.
- As stated in the text. Yes there are some damaged wires, but these damages are the same as after the manufacturing (fig 2e) and different from cracks caused by thermoshocks and can be found also far away from the area which was loaded by the transients as you noted. To further proof this statement, we introduced two images of the same area (new figure 6 which is the same sample as in fig 5b thus giving you a larger area) one after manufacturing and one after the transients. As can be seen, all damaged wires were broken during the manufacturing process and thus we will try to improve the cutting and grinding procedure although it is not a major issue as only few wires are damaged and only the tips are affected which will be sputter eroded after some time.
- No, at this stage there is no clear sign of recrystallization in pure W, as the mean temperature is not high enough. However, increasing the number of transients thus the crack depth and surface deformation, some small areas/grains at the surface can get isolated, thus overheat and then recrystallize or melt.
- Yes indeed it is known that the recrystallized state is more prone to be damaged by transients. In particular recrystallized W-wires become very brittle (we added this in the manuscript). The fact that even this “weaker” microstructure didn’t cracked further illustrates the stress reduction by using wires of 200 µm. By doing so, the mechanical load is kept in the elastic regime and thus no cracking occurs even in the recrystallized state.
- According to the MDPI authors guide, we omitted a conclusion section as it is incorporate in the short discussion section. But we extended the conclusion part there.